# Depth-Camera Based Energy Expenditure Estimation System for Physical Activity Using Posture Classification Algorithm

**DOI:** 10.3390/s21124216

**Published:** 2021-06-19

**Authors:** Bor-Shing Lin, I-Jung Lee, Chin-Shyurng Fahn, Yi-Fang Lee, Wei-Jen Chou, Meng-Luen Wu

**Affiliations:** 1Department of Computer Science and Information Engineering, National Taipei University, New Taipei City 237303, Taiwan; bslin@mail.ntpu.edu.tw (B.-S.L.); s810676105@gm.ntpu.edu.tw (I.-J.L.); s710283129@gm.ntpu.edu.tw (W.-J.C.); 2College of Electrical Engineering and Computer Science, National Taipei University, New Taipei City 237303, Taiwan; 3Department of Computer Science and Information Engineering, National Taiwan University of Science and Technology, Taipei 106335, Taiwan; M10615017@mail.ntust.edu.tw; 4Department of Computer Science and Information Engineering, Tamkang University, New Taipei City 251301, Taiwan; 158769@mail.tku.edu.tw

**Keywords:** activity classification, convolutional neural network, depth camera, energy expenditure, machine learning, multilayer perceptron, physical activity

## Abstract

Insufficient physical activity is common in modern society. By estimating the energy expenditure (EE) of different physical activities, people can develop suitable exercise plans to improve their lifestyle quality. However, several limitations still exist in the related works. Therefore, the aim of this study is to propose an accurate EE estimation model based on depth camera data with physical activity classification to solve the limitations in the previous research. To decide the best location and amount of cameras of the EE estimation, three depth cameras were set at three locations, namely the side, rear side, and rear views, to obtain the kinematic data and EE estimation. Support vector machine was used for physical activity classification. Three EE estimation models, namely linear regression, multilayer perceptron (MLP), and convolutional neural network (CNN) models, were compared and determined the model with optimal performance in different experimental settings. The results have shown that if only one depth camera is available, optimal EE estimation can be obtained using the side view and MLP model. The mean absolute error (MAE), mean square error (MSE), and root MSE (RMSE) of the classification results under the aforementioned settings were 0.55, 0.66, and 0.81, respectively. If higher accuracy is required, two depth cameras can be set at the side and rear views, the CNN model can be used for light-to-moderate activities, and the MLP model can be used for vigorous activities. The RMSEs for estimating the EEs of standing, walking, and running were 0.19, 0.57, and 0.96, respectively. By applying the different models on different amounts of cameras, the optimal performance can be obtained, and this is also the first study to discuss the issue.

## 1. Introduction

Sedentary lifestyles are common in modern society. Insufficient physical activity increases the risk of noncommunicable diseases (NCDs) [1], such as cardiovascular diseases, respiratory diseases, cancers, stroke, and diabetes. Studies have indicated that NCDs account for more than 70% of global deaths [2]. To address the problem of insufficient physical activity, physical activities must be quantified for the design of appropriate exercise plans. However, some elderly are unable to go outside to meet the basic daily requirements of physical activities because of certain healthy issues. For those elderly who can only stay at home, it is necessary to estimate their EE of indoor activities and ensure they meet the basic requirement of physical activity. Energy expenditure (EE) is one factor used for quantifying physical activities. Conventionally, people use the portable metabolic analyzer to measure EE during exercise. The common commercial models of portable metabolic analyzers include COSMED $K4b^{2}$ [3], PNOĒ, and METAMAX^®^ 3B, and the COSMED $K4b^{2}$ system is the most common system for measuring EE. Although the portable metabolic analyzers provide the most direct and accurate method for measuring calorie consumption, users of the system find it inconvenient when performing physical activities because they must carry an instrument and wear an oxygen mask during exercise.

Several studies have proposed systems using various sensors to map the EE measured by metabolic analyzers. Sensor systems can be divided into two types: contact-based [4,5,6,7] and noncontact [8,9,10,11,12]. In a contact-based sensor system, users must wear devices on body segments to measure EE. Most related research on this sensor system has adopted inertial measurement units (IMUs) for data collection. Altini et al. [4] adopted a customized platform that combined electrocardiogram sensors and accelerometers to obtain data and measure EE when users performed different physical activities in daily living. Three EE estimation methods were compared in the aforementioned study. The sensor number and the positioning of each sensor were also evaluated. Cvetković et al. [5] proposed an approach for estimating the EE of various activities in different scenarios. In addition to evaluating the EE, they analyzed the number and contributions of various sensors. Park et al. [6] evaluated the EE of six activities in daily living by combining various sensors, including an IMU and electrocardiogram sensor. Three feature selection methods were combined with the linear support vector machine (SVM) to find the best input feature set. The best feature set was tested in various models. The activity-specific approach was also adopted to improve the accuracy of EE estimation. However, the activities were only classified into two types, including static and dynamic activities. It is also not convenient to wear an IMU and electrocardiogram sensor for EE estimation. Hedegaard et al. [7] proposed an EE estimation system for seven activities during which 17 IMU sensors must be worn on body segments. Several parameters, such as gender, heart rate, acceleration, center of body weight, were input into multiple linear regression to estimate EE. Although this system can accurately estimate the EE of seven activities, wearing the clothing with 17 sensors is inconvenient. The main advantage of contact-based systems is their suitability for outdoor activities. However, wearing many sensors when performing physical activities is impractical. Sensor positioning greatly affects the accuracy of EE estimation. Moreover, the battery life of wearable devices is a critical challenge. Noncontact approaches have been proposed to solve the problems of using wearable devices to estimate EE. Kim et al. [8] developed a system based on Doppler radar to estimate the EE of upper-limb motions with a regression model. However, the average error of the aforementioned system for EE estimation was 12.75%, which is high. The other limitation of the aforementioned system is its inability to capture vertical motions, which is essential for estimating the EE of running activity. Moreover, setting a Doppler radar sensor is difficult in home settings.

Another noncontact approach for estimating EE involves a depth camera or traditional camera as a data collection tool. Yang et al. [9] proposed an EE estimation system based on smartphone cameras for workouts. The system can accurately count the repetitions and estimate the EE of four types of workouts: sit-ups, push-ups, jumping jacks, and squats. However, the system provides quantification for only the four aforementioned workouts, which are uncommon in daily living. Koporec et al. [10] developed a noncontact method for EE estimation by using a camera and classical image processing approaches. However, the error of EE estimation in the aforementioned study was high. With the development of artificial intelligence, deep learning approaches have also been adopted for EE estimation. Na et al. [11] proposed a deep-learning-based method for estimating the EE of different levels of physical activities. A convolutional neural network (CNN)-based method was used in their study for EE prediction [13]. The aforementioned research is representative of the applications of deep learning algorithms to EE estimation; however, the EEs of only light physical activities were estimated in the aforementioned study.

Lin et al. [12] proposed an EE estimation system by using Kinect depth cameras. The cameras were installed at three locations, namely the rear view, rear side view, and side view. Moreover, five regression-based models were compared to determine the model with the highest accuracy. Lin et al. estimated the EEs of only moderate-to-vigorous activities. Light activities, such as standing, were not considered; however, standing is an everyday physical activity. Another limitation of the aforementioned study was the high computational time required because of the complicated feature extraction process.

The proposed study can be regarded as an extension of the study of Lin et al. The motivation of the proposed study is to propose a Kinect-based EE estimation system to solve the problems and limitations associated with the previous research of Lin et al. This study investigated if applying the physical activity classification algorithm to the EE estimation system can improve the accuracy of EE estimation, which has not been confirmed in previous studies. Moreover, the proposed study investigated the best model for each type of physical activity for different camera settings. The study adopted a physical activity recognition algorithm to classify the postures captured by Kinect depth cameras into three activities, namely standing, walking, and running. Only 18 velocity-based features were used for EE estimation in this study to lower the computational complexity for feature extraction and selection. Three types of models were adopted to develop an independent model for each activity, and the model with the best performance for each activity was selected. The proposed system hypothesized that using the posture classification algorithm can improve the accuracy of EE estimation and is expected to obtain the optimal model for each physical activity for different camera settings.

## 2. Methodology

Figure 1 depicts the system architecture used in this study. The system was used to perform five tasks: data acquisition, data preprocessing, feature extraction, physical activity classification, and EE estimation. The performance of three models, namely linear regression (LR), multilayer perceptron (MLP), and convolutional neural network (CNN) models, for estimating the EE of each activity were compared. After physical activity classification, the independent models for activity prediction were used for training and EE estimation.

### 2.1. Data Acquisition

The dataset used in this study was obtained from a study conducted in 2019 [12]. Three XBOX 360 Kinect cameras (Microsoft Corp., Redmond, WA, USA) were placed along three directions, namely the side, rear side, and rear view directions. The positions of the Kinect cameras are illustrated in Figure 2. Each Kinect camera was installed and fixed to a tripod 0.9 m high. The distance between each camera and the participant was set to approximately 2 m, which ensured that the Kinect cameras could capture the participant’s entire body. The sampling frequency of Kinect is 30 Hz.

During experiments, participants were asked to wear a portable metabolic analyzer ($K4b^{2}$, COSMED, Rome, Italy) to record the pulmonary gas exchange by breath with an accuracy of deviation of ±0.02% O_2_ and ± 0.01% CO_2_.

A total of 21 subjects (10 men and 11 women) were recruited for the experiment. Their gender, ages, weights, and body fat ratios are listed in Table 1. The mean and standard deviation (SD) of the age of the participants is 21.90 ± 1.55 years old. The mean and SD of the weight is 60.20 ± 7.60 kg. The mean and SD of the body fat rate is 20.69 ± 7.37%.

People with infectious or chronic diseases were excluded from this study. All participants provided informed consent before participating in the experiments. The experimental procedures used in this study were approved by the Institutional Review Board (IRB) of Cathay General Hospital, Taipei, Taiwan (IRB code: CGH-NTPU105001).

Experimental details for each activity are listed in Table 2. Various physical activities, including standing, walking, and running at various speeds, were included in the experiments. Standing, walking, and running are regarded as the light, moderate, and vigorous activities in this study. The walking speeds considered were 4.8, 5.6, and 6.4 km/h, and the running speeds considered were 8.0 and 8.3 km/h and were set on the treadmill. Participants performed each activity for 5 min on a treadmill. In the pilot study, a treadmill was used instead of free-living conditions because a stable movement speed can be maintained with the treadmill. The resting times between each light-to-moderate activities and moderate-to-vigorous activities were 5 and 10 min, respectively.

The skeleton sequential data were collected with Kinect cameras while the participants were performing the physical activity on the treadmill. The first 3 min of each activity represented a non-steady state, and data from this period were discarded in this study [14]. Two minutes of steady-state data were used for physical activity classification and EE prediction.

To develop an EE estimation model, the expenditure data for each physical activity should be collected as the ground truth. In this study, the metabolic equivalent of task (MET) was regarded as the measurement unit. The MET is calculated as the ratio of the rate of energy expended during physical activity to the mass of a person. This ratio is used by many aerobic training organizations to estimate exercise intensity and EE.

### 2.2. Data Preprocessing

Data preprocessing can be divided into two parts: coordinate system transformation and noise reduction. Coordinate system transformation aims to transform the reference coordinate of each skeletal value into the body center. The initial reference coordinate of skeletal values was the center of a Kinect camera, but the varying object distances from a Kinect camera would cause problems of action recognition. For example, when the participants were walking or running on the treadmill, their bodies moved back and forth, which caused unstable variations in the skeletal data. According to the previous study, the referenced coordinate should be translated from the Kinect camera to the origin of body center, which is the shoulder center [15]. In this study, the reference coordinate was set to the center of the shoulder as the previous study had suggested. The reference coordinate of other joints was transformed by using the relative distance between the joint and the shoulder center. The formula for coordinate system transformation can be expressed as follows:(1)$$\left(x_{i}{}^{\text{′}},\;y_{i}{}^{\text{′}},\;z_{i}{}^{\text{′}}\right)=\left(x_{i}-x_{r},\;y_{i}-y_{r},\;z_{i}-z_{r}\right),$$
where the coordinate of the shoulder center is $\left(x_{r},y_{r},z_{r}\right)$, i=1, …, j is the index of the joint, j is the total selected number of joints, which is six and will be explained in the following section, and $\left(x_{i}{}^{\prime },y_{i}{}^{\prime },z_{i}{}^{\prime }\right)$ denotes the skeleton coordinates after transformation.

Kinect cameras are sensitive to backgrounds, light, and surrounding objects. Noise occurs during skeletal tracking [16]. Therefore, a moving average filter, which averages the coordinates of two previous and two subsequent frames [17], can be used to remove the noise. The formula for applying the moving average filter is:(2)$$\left(x_{k}{}^{*},\;y_{k}{}^{*},\;z_{k}{}^{*}\right)=\frac{1}{5}\left(\overset{k+2}{\underset{n=k-2}{\sum }}x_{n}{}^{\text{′}},\;\overset{k+2}{\underset{n=k-2}{\sum }}y_{n}{}^{\text{′}},\;\overset{k+2}{\underset{n=k-2}{\sum }}z_{n}{}^{\text{′}}\right)$$
where k represents the current frame, k−2 to k+2 represent two previous frames and two subsequent frames, and $\left(x_{k}{}^{*},\;y_{k}{}^{*},\;z_{k}{}^{*}\right)$ is the new skeleton coordinate after processing with the moving average filter.

### 2.3. Feature Extraction

Because the speed of each activity is varying, the mean velocity between every two consecutive frames might be an important factor to discriminate the activities. The formula is presented in (3). The velocity of a joint is calculated by obtaining the difference of velocity between two consecutive frames [18]. $v_{x}^{k},\;v_{y}^{k}$, and $v_{z}^{k}$ means the mean velocity in *x*, *y*, and *z* directions and will be used to train the models. The accuracy of the positions of joints from Kinect was also discussed and validated in the previous study to prove that the data used for this study are reliable [19].
(3)$$\left(v_{x}^{k},\;v_{y}^{k},v_{z}^{k}\right)=\frac{\left(x_{k}{}^{*},\;y_{k}{}^{*},\;z_{k}{}^{*}\right)-\left(x_{k-1}{}^{*},\;y_{k-1}{}^{*},\;z_{k-1}{}^{*}\right)}{t_{k}-t_{k-1}}$$

According to the related research [20], treadmill workouts, such as standing, walking, and running, involve periodic and symmetric movements. Thus, to reduce the computational complexity, only half of the joints in the body were used to analyze physical activities. The six joints selected in this study were the shoulder, elbow, wrist, hip, knee, and ankle on the left side (red circles in Figure 3). The six selected joints were imaged in three directions simultaneously by using the three cameras. Three-dimensional data were obtained for each joint. Therefore, 18 velocities were obtained as features for further analysis.

Principal component (PC) analysis (PCA) is used to reduce feature dimensionality. In this study, PCs that cumulatively accounted for more than 90% of the overall variances were retained [21]. The numbers of feature subsets from the side, rear side, and rear views were 10, 9, and 7, respectively. The cumulative explained variance of the retained PCs is plotted in Figure 4.

### 2.4. Physical Activity Classification

To improve the accuracy of EE estimation, physical activities were classified as standing, walking, or running. An SVM algorithm was adopted to classify physical activities [22]. SVM is a widespread method to solve the classification problems. It can convert the raw data from lower to higher dimension using the kernel functions. After converting to higher dimension, the optimal hyper plane can be found to separate the data from different classes with the maximized margin. The advantage of SVM is that it can solve linear inseparable problems after projecting the raw data to a higher dimension. In this study, three SVM classifiers were built to classify the physical activities into three activities with one-against-all approach, and radial basis function kernel was used as the kernel function.

### 2.5. EE Prediction

To obtain the optimal model for estimating the EE during physical activities, the estimation results obtained by using three models, namely LR, MLP, and CNN, were compared. The reason of testing the three models is that these are the most common models in statistical, machine learning, and deep learning fields. The LR model can be used to determine whether the ground truth and estimated output are highly correlated. The MLP model is a type of feedforward artificial neural network, which comprises an input layer, an output layer, and several hidden layers [23]. The MLP model used in this study contained five layers: one input layer, three hidden layers, and one output layer, as presented in Figure 5. Different numbers of nodes *n* were tested to obtain the optimal number of neurons in the hidden layers to achieve optimal regression results. The logistic function was used as the activation function to estimate EE for different numbers of neurons, and mean square error (MSE) was used to evaluate the performance of different numbers of nodes. The number of nodes varied from 10 to 90 in increments of 10 neurons.

CNN is widely used to recognize objects and patterns in images [24]. In this study, CNN was used to automatically extract the features of kinematic data. The structure of the constructed CNN model is depicted in Figure 6. The CNN model used in this study comprised one input layer, two convolutional layers, two max-pooling layers, and one dense layer. The kernel size for each convolutional layer was 3 × 3. The rectified linear unit activation function was used in each convolutional layer [25]. Eighteen velocity data were input as the 18×1 input layer. The two convolutional layers were used to detect the features of the input data and create the feature maps automatically. The fully connected layers comprised two hidden layers with 16 hidden units per layer. In addition, the final output layer will output the result of the MET.

To evaluate the performance of EE estimation and compare the performance with the related research, three indicators used in the related research [5,6,7,10,12], mean absolute error (MAE), MSE, and root MSE (RMSE), were adopted to evaluate the performance. The formulas for calculating MAE, MSE, and RMSE are as follows:(4)$$\mathrm{MAE}=\frac{1}{N}\overset{N}{\underset{k=1}{\sum }}\left|{\hat{Y}}_{k}-Y_{k}\right|$$
(5)$$\mathrm{MSE}=\frac{1}{N}\overset{N}{\underset{k=1}{\sum }}{\left({\hat{Y}}_{k}-Y_{k}\right)}^{2}$$
(6)$$\mathrm{RMSE}=\sqrt{\overset{N}{\underset{k=1}{\sum }}\frac{{\left({\hat{Y}}_{k}-Y_{k}\right)}^{2}}{N}}$$
where ${\hat{Y}}_{k}$ denotes the estimated EE from each model, and $Y_{k}$ denotes the actual EE from $K4b^{2}$ portable calorimetry system. *k* = 1, …, *N* is the index of the data point. *N* is the amount of the data points. The unit for the MAE and RMSE was the MET. The ranges for MAE, MSE, and RMSE are [0, ∞].

Ten-fold cross-validation was used to evaluate the accuracy of different EE predictive models [26]. The final accuracy of each model was the average accuracy obtained through 10-fold cross-validation.

## 3. Results

For the MLP model, different nodes were tested to obtain the optimal model for the further comparisons. Figure 7 indicated that the lowest MSE was achieved when 70 neurons per hidden layer were used. The MLP with three hidden layers which contains 70 neurons per layer was adopted for the further comparisons.

The performance evaluation of the models was divided into three parts. The EE estimation performance of (i) a general model for all physical activities, (ii) the physical activity classification method, and (iii) independent models for different physical activities were evaluated.

### 3.1. EE Estimation Performance of the General Model

A general model was built for estimating the energy expenditure of all physical activities for two purposes. The first is for comparison with independent models for physical activity classification. The second is that if only one Kinect camera is available, the optimal position of the general model can be used.

Data for different physical activities were used to evaluate the EE estimation performance of the LR, MLP, and CNN models. The performance of the models with and without PCA was also examined to evaluate whether PCA improves the performance of the general model. The performance of the models with and without PCA is presented in Table 3, Table 4 and Table 5. According to Table 3, Table 4 and Table 5, when data from all physical activities were used, the models without PCA outperformed those with PCA. The MLP and CNN models exhibited smaller errors than did the LR model. Moreover, the smallest estimation error was obtained when a Kinect camera was set along the side view and the MLP model without PCA was adopted. The MAE, MSE, and RMSE under the aforementioned settings were 0.55, 0.66, and 0.81, respectively.

### 3.2. Accuracy of Physical Activity Classification

Because the EE estimation method proposed in this study is mainly based on physical activity classification, the accuracy of physical activity classification must be examined. The performance of physical activity classification with and without PCA is presented in Table 6.

The performance differences in physical activity classification with and without PCA were nonsignificant. Moreover, the accuracy of physical activity classification without PCA was higher than that with PCA. Accuracy without PCA could reach 99.55%.

Therefore, PCA was not adopted for further comparisons of the models and experimental settings for different activities. 

### 3.3. EE Estimation Performance with Physical Activity Classification

Table 7, Table 8 and Table 9 indicate the EE estimation performance with physical activity classification. The smallest error in the EE estimation for standing was achieved when a Kinect camera was installed along the rear view and the CNN model was used. The MAE, MSE, and RMSE of the EE estimation for standing were 0.15, 0.04, and 0.19, respectively. The smallest error in the EE estimation for walking was achieved when a Kinect camera was set along the side view and the CNN model was used. The MAE, MSE, and RMSE of the EE estimation for walking were 0.45, 0.33, and 0.57, respectively. The smallest error in the EE estimation for running was obtained when a Kinect camera was set along the rear view and the MLP model was used. The MAE, MSE, and RMSE of the EE estimation for running were 0.66, 0.94, and 0.96, respectively.

## 4. Discussion

In the proposed method, physical activity classification is performed before the EE estimation model is developed. To compare with the performance in different models and experimental settings, the performance evaluation was divided into three parts. First, the performance of a general model for all physical activities was investigated. According to the results in Table 3, Table 4 and Table 5, the smallest error in EE was obtained when the Kinect camera was set along the side view and the MLP model was used. The MAE, MSE, and RMSE in the EE under the aforementioned settings were 0.55, 0.66, and 0.81, respectively. Thus, when only one Kinect camera is available, optimal EE estimation performance is obtained when the camera is set along the side view and the MLP model is used. For an individual who weighs 60 kg and exercises for 1 h, the error between the EE estimated with the general model and the actual EE is 33 kcal.

Second, the accuracy of physical activity classification was examined with and without PCA. The classification accuracies with and without PCA were examined because the primary aim of this study was to determine whether physical activity classification improves EE estimation accuracy. Therefore, evaluating the accuracy of physical activity classification was crucial. As presented in Table 6, the accuracy of EE estimation along the side view was 99.55% without PCA and 98.49% with PCA. Therefore, superior EE estimation performance was obtained without PCA. A possible reason for this finding is the loss of velocity characteristics after using PCA because PCA results in the projection of data from high to low dimensions. Although the accuracy of using PCA is lower in this study, it was still helpful while collecting a larger dataset in future works because larger dataset will increase the computational cost of the model, and PCA can reduce the complexity of classification problems and make the predictive model more stable.

Third, the performance of EE estimation with physical activity classification was investigated. Table 7, Table 8 and Table 9 present the EE estimation performance of different models under different experimental settings for three physical activities. The results indicated that the CNN model provided the highest EE estimation accuracy for light and moderate-intensity activities, such as standing and walking. The MLP model exhibited the highest EE estimation accuracy for vigorous activities such as running. The optimal setting for achieving optimal EE estimation performance with physical activity classification is illustrated in Figure 8. The optimal setting involves two Kinect cameras along the side and rear views as well as a combination of MLP and CNN models. After physical activity classification, if the physical activity is classified as standing, walking, or running, then the EE is estimated using the CNN model for standing, CNN model for walking, or MLP model for running, respectively. The combination of various models reduced computational complexity because fewer data were required for independent model training than for general model training.

The data from the rear side view did not provide optimal results for any experimental setting possibly because the data provided by the camera on the rear side view have already been projected on the other two planes. Therefore, the skeletal data from the Kinect cameras along only the side and rear views should be used in future studies. Moreover, the results show that after applying physical activity classification for EE estimation, the performance of estimating EE for light to moderate activity, such as standing and walking, will be improved; however, the performance for vigorous activity will not be improved by applying physical activity classification according to the results.

Table 10 presents a comprehensive comparison of the proposed method with methods presented in related studies. Several criteria, such as sensors, accuracy, indoor/outdoor usage, limitation of battery time, and requirement of wearable devices, are discussed among all the listed studies in Table 10. The proposed method can estimate the EE of light-to-vigorous activities. Only Kinect cameras were used in this study. Of all the compared methods, only the proposed method does not require personal data, such as weight, heart rate, and gender. The proposed method does not have the problem of battery life but can be used for indoor activities only. None of the compared camera-based systems require wearable devices. The accuracy of the proposed general model is the same as that of the system proposed by Cvetković et al. [5] but lower than that of the system proposed by Lin et al. [12]. However, in the EE estimation for standing and walking, the proposed system outperforms that of Lin et al. This result is acceptable because standing and walking account for most of the exercising time. The proposed method only uses velocity-based features (only 18 features) for EE estimation. Thus, the computational complexity of the proposed method is lower than that of other methods.

This study has certain limitations. First, not all models considered in this study were fine tuned to their optimal parameters. In this study, only the neurons of the hidden layers in the MLP model were fine tuned. Second, two Kinect cameras are still required to obtain the best EE prediction performance. However, a practical solution requiring the use of only one Kinect camera would be more appropriate.

To solve the aforementioned limitations, in the future studies, the optimal parameters for all models will be investigated. Deep learning approaches for sequential datasets, such as the algorithms based on recurrent neural networks [27,28] and long short-term memory networks [29], will also be considered for EE estimation. Moreover, it is necessary to build a robust model with only one Kinect camera for more practical usage. The approaches for converting the coordinates of a Kinect camera from one direction to other direction will be considered for improving the performance and efficiency of the EE estimation model on any direction.

## 5. Conclusions

This paper proposed an EE estimation system based on physical activity classification. Three depth cameras were set at three locations, namely the side, rear side, and rear views, to obtain the kinematic data and test the performance of EE estimation. Three EE estimation models, including LR, MLP, and CNN models, were compared and determined the model with optimal performance in varying experimental settings.

The experimental results indicated that when only one camera is available, the optimal solution involves setting the camera along the side view and estimating EE with the MLP model. The MAE, MSE, and RMSE of the EE under the aforementioned settings were 0.55, 0.66, and 0.81, respectively. In physical activity classification, the highest EE estimation accuracy for light-to-moderate activities, such as standing and walking, was obtained when Kinect cameras were set along the side and rear views and the CNN model was used. The highest EE estimation accuracy for vigorous activities, such as running, was obtained when Kinect cameras were set along the side and rear views and the MLP model was used. The RMSEs for estimating the EEs of standing, walking, and running were 0.19, 0.57, and 0.96, respectively. The camera on the rear side view did not yield optimal performance in all the experimental settings in this study. The results indicated that after applying physical activity classification, the performance of estimating EE of standing and walking was improved compared with that of the general model; however, the performance of running was not improved after applying physical activity classification. By using the proposed system, the EE of physical activities can be accurately estimated.

The main innovation of this research is that this is the first study to discuss the performance of EE estimation in different experimental settings, including one and two cameras, and the speed of different types of activities. With the proposed system, the subject can choose the best setting depending on their exercising environments to estimate EE.

## Figures and Tables

**Figure 1 sensors-21-04216-f001:**
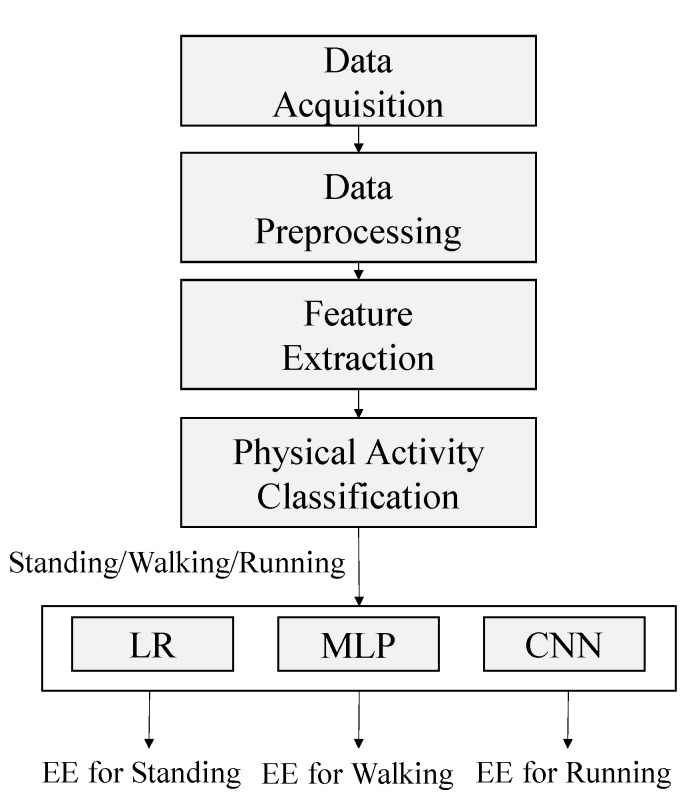
System architecture used in the proposed method.

**Figure 2 sensors-21-04216-f002:**
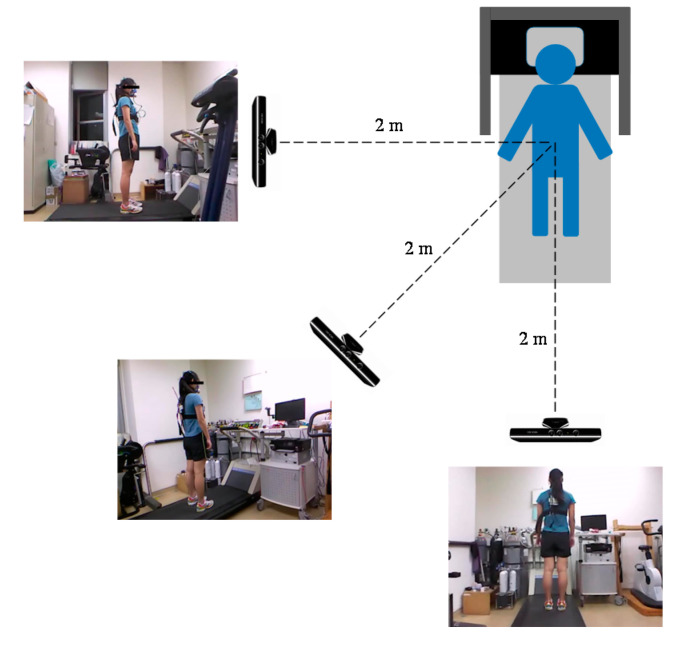
Locations of the Kinect cameras.

**Figure 3 sensors-21-04216-f003:**
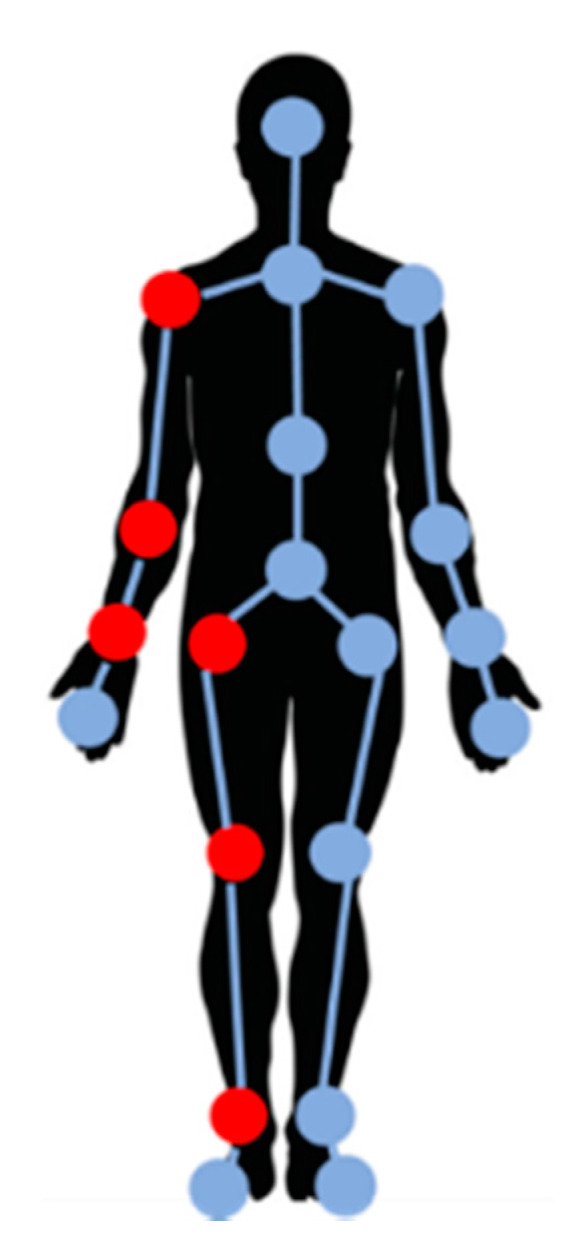
Map of the Kinect-tracked skeleton joints. The red-marked joints represent the selected joints.

**Figure 4 sensors-21-04216-f004:**
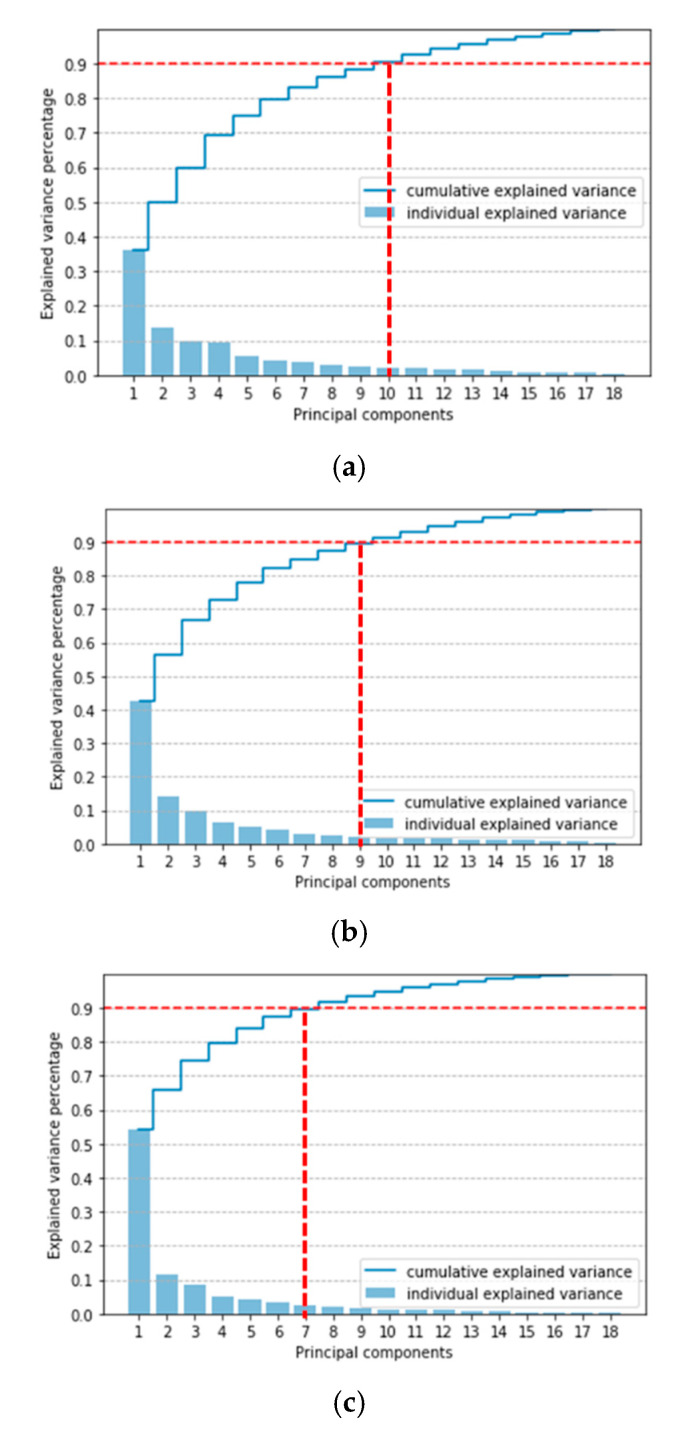
Cumulative and individual explained variances of the retained PCs for each Kinect placement: (**a**) side view, (**b**) rear side view, and (**c**) rear view.

**Figure 5 sensors-21-04216-f005:**
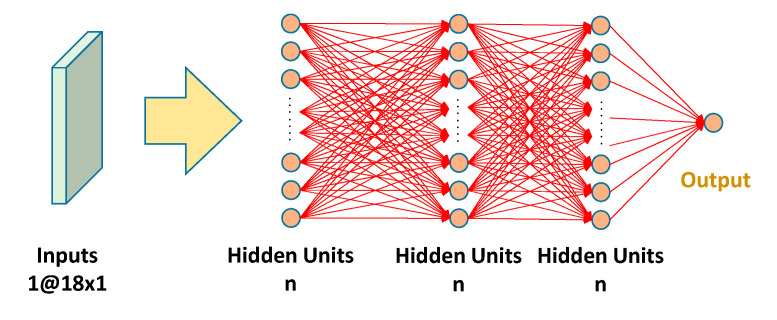
Structure of the constructed MLP model.

**Figure 6 sensors-21-04216-f006:**
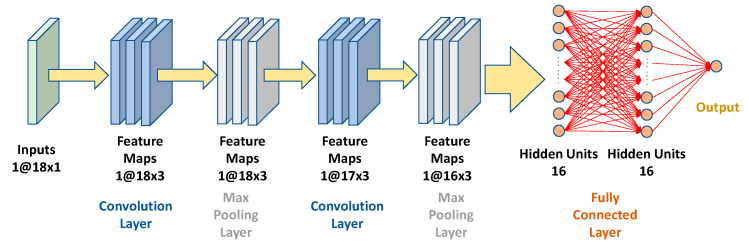
Structure of the constructed CNN model.

**Figure 7 sensors-21-04216-f007:**
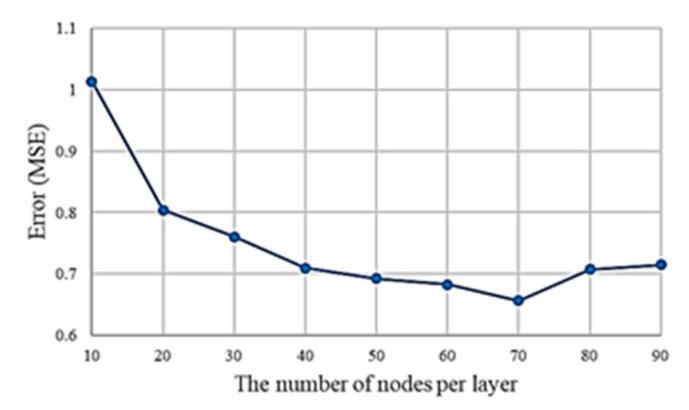
MSEs for different numbers of nodes (from 10 to 90) in the MLP model.

**Figure 8 sensors-21-04216-f008:**
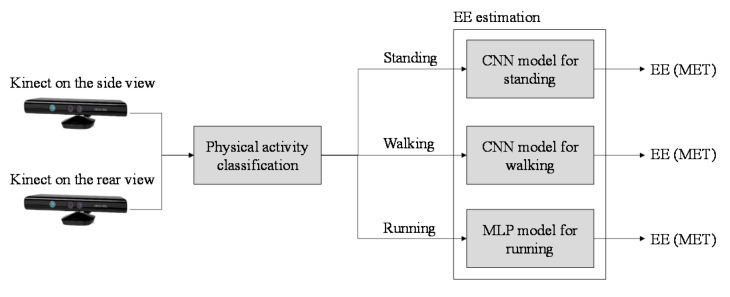
Optimal setting of the proposed system.

**Table 1 sensors-21-04216-t001:** Participants’ data.

Number of Subjects	21
Sex	10 males, 11 females
Age (years old)	21.90 ± 1.55
Weight (kg)	60.20 ± 7.60
Body Fat Rate (%)	20.69 ± 7.37

**Table 2 sensors-21-04216-t002:** Experimental details of each activity.

Type	Activity	Speed (km/h)	Time (min)	Rest Time (min)
Light	Standing	0	5	0
Moderate	Walking	4.8	5	5
	Walking	5.6	5	5
	Walking	6.4	5	5
Vigorous	Running	8.0	5	10
	Running	8.3	5	10

**Table 3 sensors-21-04216-t003:** LR model results obtained with and without PCA.

	Side		Rear Side		Rear	
	Without PCA	With PCA	Without PCA	With PCA	Without PCA	With PCA
MAE (MET)	1.29	1.38	1.51	1.54	1.25	1.50
MSE (MET^2^)	2.94	3.46	3.85	4.04	2.62	3.69
RMSE (MET)	1.71	1.86	1.96	2.01	1.62	1.92

**Table 4 sensors-21-04216-t004:** MLP model results obtained with and without PCA.

	Side		Rear Side		Rear	
	Without PCA	With PCA	Without PCA	With PCA	Without PCA	With PCA
MAE (MET)	0.55	0.67	0.85	1.06	0.61	0.82
MSE (MET^2^)	0.66	1.12	1.65	2.52	0.81	1.52
RMSE (MET)	0.81	1.06	1.28	1.58	0.90	1.23

**Table 5 sensors-21-04216-t005:** CNN model results obtained with and without PCA.

	Side		Rear Side		Rear	
	Without PCA	With PCA	Without PCA	With PCA	Without PCA	With PCA
MAE (MET)	0.63	0.78	0.92	1.08	0.70	0.89
MSE (MET^2^)	0.81	1.30	1.66	2.36	0.95	1.60
RMSE (MET)	0.90	1.14	1.29	1.53	0.98	1.26

**Table 6 sensors-21-04216-t006:** Accuracy of the SVM algorithm with and without PCA for classifying different activities at three Kinect camera locations.

	Side	Rear Side	Rear
Without PCA	99.55%	96.33%	98.70%
With PCA	98.49%	92.16%	95.72%

**Table 7 sensors-21-04216-t007:** EE estimation performance of independent LR models for different activities.

	Side			Rear Side			Rear		
	Standing	Walking	Running	Standing	Walking	Running	Standing	Walking	Running
MAE (MET)	0.24	0.70	1.13	0.22	0.75	1.23	0.20	0.62	1.12
MSE (MET^2^)	0.09	0.82	2.26	0.08	0.90	2.55	0.06	0.62	2.11
RMSE (MET)	0.30	0.90	1.50	0.28	0.95	1.60	0.25	0.78	1.45

**Table 8 sensors-21-04216-t008:** EE estimation performance of independent MLP models for different activities.

	Side			Rear Side			Rear		
	Standing	Walking	Running	Standing	Walking	Running	Standing	Walking	Running
MAE (MET)	0.18	0.44	0.69	0.17	0.59	0.88	0.16	0.47	0.66
MSE (MET^2^)	0.06	0.34	0.96	0.05	0.59	1.64	0.04	0.41	0.94
RMSE (MET)	0.24	0.58	0.98	0.23	0.77	1.28	0.20	0.64	0.96

**Table 9 sensors-21-04216-t009:** EE estimation performance of independent CNN models for different activities.

	**Side**			**Rear Side**			**Rear**		
	**Standing**	**Walking**	**Running**	**Standing**	**Walking**	**Running**	**Standing**	**Walking**	**Running**
MAE (MET)	0.16	0.45	0.76	0.17	0.56	0.87	0.15	0.47	0.73
MSE (MET^2^)	0.04	0.33	1.18	0.05	0.54	1.43	0.04	0.37	0.98
RMSE (MET)	0.21	0.57	1.08	0.22	0.74	1.19	0.19	0.60	0.99

**Table 10 sensors-21-04216-t010:** Comparison of the proposed method with methods proposed in related studies.

	Cvetković et al. [5]	Park et al. [6]	Koporec et al. [10]	Lin et al. [12]	Hedegaard et al. [7]	The Proposed Method
Year	2016	2017	2018	2019	2020	2020
Activity	Light to vigorous	Light to vigorous	Light to vigorous	Moderate to vigorous	Light to vigorous	Light to vigorous
Sensors	Multi-sensors	Multi-sensors	Camera	Kinect camera	IMU sensors	Kinect camera
Personal Data	Yes	Yes	No	Yes	Yes	No
Indoor/Outdoor	Both	Both	Indoor	Indoor	Both	Indoor
Limitation of battery time	Yes	Yes	No	No	Yes	No
Wearable devices are needed	Yes	Yes	No	No	Yes	No
RMSE (MET)	0.81	0.89	2.22	0.76	Sedentary—0.29	General model—0.81
						Standing—0.19
					Dynamic—1.14	Walking—0.57
						Running—0.96

## Data Availability

No new data were created or analyzed in this study. Data sharing is not applicable to this article.

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
