# Peer review of "Depth-Camera Based Energy Expenditure Estimation System for Physical Activity Using Posture Classification Algorithm"

_sensors, 2021, doi:10.3390/s21124216_

Round 1
Reviewer 1 Report
Recommendation: the author should prepare a minor revision.
The contribution of this paper is fair. However, there are also some comments which might help the authors to improve this paper.
In “Introduction”, the thirteenth line of the first paragraph. “In Section IV, the results after the model prediction will be presented. In Section V, the experimental results will be discussed. In Section VI, the conclusions of the research are described.”, Three successive sentences begin with the same word. Consider rewording the sentence or use a thesaurus to find a synonym.
Some sentences in the article have tense problems, the following are the areas that need improvement:
- In “Abstract ”, the tenth line of the first paragraph. “The results shown that if only one depth camera is available, optimal EE estimation can be obtained using the side view and MLP model.”, “shown” may be “have shown” .
- In “4 Results”, the first line of the first paragraph. “The EE estimation performance of (i) a general model for all physical activities, (ii) the physical activity classification method, and (iii) independent models for different physical activities was evaluated.”, “was” may be “were” .
- In “5 Discussion”, the first line of the first paragraph. “The data from the rear side view did not provide optimal results for any experimental setting possibly because the data provided by the camera on the rear side view has already projected on the other two planes.”, “projected” may be “been projected”.
The subject of some sentences in the passage does not seem to agree with the predicate noun in number, the following are the areas that need improvement:
- In “Abstract ”, the sixteenth line of the first paragraph. “By applying the different models on different amount of cameras, the optimal performance can be obtained, and this is also the first study to discuss the issue.”, “amount” may be “amounts”.
- In “3.1 Data Acquisition”, the seventh line of fifth paragraph. “The resting times between each light‐to‐moderate activity and moderate‐to‐vigorous activity were 5 and 10 min, respectively.”, “activity” may be “activities”.
- In “4.3 EE Estimation Performance with Physical Activity Classification”, the first line of first paragraph. “Tables VII to IX indicate the EE estimation performance with physical activity classification.”, “indicate” may be “indicates”.
- In “5 Discussion”, the second line of third paragraph. “Tables VII to IX present the EE estimation performance of different models under different experimental settings for three physical activities.”, “present” may be “presents”.
- In “6 Conclusion”, the fifth line of fifth paragraph. “The approaches for converting the coordinates of a Kinect camera from one direction to other directions will be considered for improving the performance and efficiency of the EE estimation model on any directions.”, “directions” may be “direction”
Some sentences in the passage use the article "an" or "a" incorrectly, the following are the areas that need improvement:
- In “Related work”, the first line of the fourth paragraph. “The motivation of the proposed study is to propose an Kinect‐based EE estimation system to solve the problems and limitations associated with the previous research of Lin et al..”, “an Kinect‐based” may be “a Kinect‐based”.
- In “3.4 Physical Activity Classification”, the second line of first paragraph. “A SVM algorithm was adopted to classify physical activities [22].”, “A SVM algorithm” may be “An SVM algorithm”
In “Related work”, the first line of the fourth paragraph. “The motivation of the proposed study is to propose an Kinect‐based EE estimation system to solve the problems and limitations associated with the previous research of Lin et al..”, there are two consecutive points in this sentence. Consider deleting one.
In “3.1 Data Acquisition”, the third line of seventh paragraph. “The MET is calculated as the ratio of the rate of energy expended during a physical activity to the mass of a person.”, the indefinite article, a, may be redundant when used with the uncountable noun activity in your sentence. Consider removing it.
In “3.2 Data Preprocessing”, the second line of first paragraph. “The aim of coordinate system transformation is to transform the reference coordinate of each skeletal value into the body center.”, it may be “coordinate system transformation aims to transform the reference coordinate of each skeletal value into the body center.”
In “6 Conclusion”, the tenth line of second paragraph. “The camera on the rear side view did not yield optimal performance in any experimental setting.”, the word “any” doesn't seem to fit this context. Consider replacing it with a different one
Generally, there is something offered in the paper, but the authors should prepare a minor revision for the paper.
Author Response
Author’s replies to the comments of Reviewer 1:
[Comment 1]
In “Introduction”, the thirteenth line of the first paragraph. “In Section IV, the results after the model prediction will be presented. In Section V, the experimental results will be discussed. In Section VI, the conclusions of the research are described.”, Three successive sentences begin with the same word. Consider rewording the sentence or use a thesaurus to find a synonym.
Answer:
Many thanks for the reviewer’s comments. Because of the suggestion from the other reviewer, we have removed the following sentences in Introduction: “The paper is organized as follows. In Section II, the related works will be reviewed. In Section III, the proposed methods for predicting energy expenditure will be introduced. In Section IV, the results after the model prediction will be presented. In Section V, the experimental results will be discussed. In Section VI, the conclusions of the research are described.”
[Comment 2]
Some sentences in the article have tense problems, the following are the areas that need improvement:
In “Abstract ”, the tenth line of the first paragraph. “The results shown that if only one depth camera is available, optimal EE estimation can be obtained using the side view and MLP model.”, “shown” may be “have shown” .
In “4 Results”, the first line of the first paragraph. “The EE estimation performance of (i) a general model for all physical activities, (ii) the physical activity classification method, and (iii) independent models for different physical activities was evaluated.”, “was” may be “were” .
In “5 Discussion”, the first line of the first paragraph. “The data from the rear side view did not provide optimal results for any experimental setting possibly because the data provided by the camera on the rear side view has already projected on the other two planes.”, “projected” may be “been projected”.
Answer:
Many thanks for the reviewer’s comments. We have modified all of the tense problems reported by the reviewer and the highlighted the modified parts.
[Comment 3]
The subject of some sentences in the passage does not seem to agree with the predicate noun in number, the following are the areas that need improvement:
In “Abstract ”, the sixteenth line of the first paragraph. “By applying the different models on different amount of cameras, the optimal performance can be obtained, and this is also the first study to discuss the issue.”, “amount” may be “amounts”.
In “3.1 Data Acquisition”, the seventh line of fifth paragraph. “The resting times between each light‐to‐moderate activity and moderate‐to‐vigorous activity were 5 and 10 min, respectively.”, “activity” may be “activities”.
In “4.3 EE Estimation Performance with Physical Activity Classification”, the first line of first paragraph. “Tables VII to IX indicate the EE estimation performance with physical activity classification.”, “indicate” may be “indicates”.
In “5 Discussion”, the second line of third paragraph. “Tables VII to IX present the EE estimation performance of different models under different experimental settings for three physical activities.”, “present” may be “presents”.
In “6 Conclusion”, the fifth line of fifth paragraph. “The approaches for converting the coordinates of a Kinect camera from one direction to other directions will be considered for improving the performance and efficiency of the EE estimation model on any directions.”, “directions” may be “direction”
Answer:
Many thanks for the reviewer’s comments. We have modified all of the grammar errors which the reviewer listed and highlighted all the modified words.
The sentences mentioned by the reviewer in the conclusion have been moved to the Discussion section.
[Comment 4]
Some sentences in the passage use the article "an" or "a" incorrectly, the following are the areas that need improvement:
In “Related work”, the first line of the fourth paragraph. “The motivation of the proposed study is to propose an Kinect‐based EE estimation system to solve the problems and limitations associated with the previous research of Lin et al..”, “an Kinect‐based” may be “a Kinect‐based”.
In “3.4 Physical Activity Classification”, the second line of first paragraph. “A SVM algorithm was adopted to classify physical activities [22].”, “A SVM algorithm” may be “An SVM algorithm”
Answer:
Many thanks for the reviewer’s comments. We have modified all of the mistakes of “an” or “a” which were suggested by the reviewer and highlighted all of the modified words.
The Related work section have been merged into Introduction section.
[Comment 5]
In “Related work”, the first line of the fourth paragraph. “The motivation of the proposed study is to propose an Kinect‐based EE estimation system to solve the problems and limitations associated with the previous research of Lin et al..”, there are two consecutive points in this sentence. Consider deleting one.
Answer:
Many thanks for the reviewer’s comments. We have deleted one point in this sentence and highlighted the remained point.
The Related work section have been merged into Introduction section.
[Comment 6]
In “3.1 Data Acquisition”, the third line of seventh paragraph. “The MET is calculated as the ratio of the rate of energy expended during a physical activity to the mass of a person.”, the indefinite article, a, may be redundant when used with the uncountable noun activity in your sentence. Consider removing it.
Answer:
Many thanks for the reviewer’s comments. We have removed “a” in the sentence and highlighted the sentence.
[Comment 7]
In “3.2 Data Preprocessing”, the second line of first paragraph. “The aim of coordinate system transformation is to transform the reference coordinate of each skeletal value into the body center.”, it may be “coordinate system transformation aims to transform the reference coordinate of each skeletal value into the body center.”
Answer:
Many thanks for the reviewer’s comments. We have changed the sentence into “Coordinate system transformation aims to transform the reference coordinate of each skeletal value into the body center.” as the reviewer suggested and highlighted it.
[Comment 8]
In “6 Conclusion”, the tenth line of second paragraph. “The camera on the rear side view did not yield optimal performance in any experimental setting.”, the word “any” doesn't seem to fit this context. Consider replacing it with a different one.
Answer:
Many thanks for the reviewer’s comments. We have changed ”any experimental setting” into “all the experimental settings in this study” and highlighted the modified part.
[Comment 9]
Generally, there is something offered in the paper, but the authors should prepare a minor revision for the paper.
Answer:
Many thanks for the reviewer’s comments. We have modified all the sentences and words suggested by the reviewer and highlighted them in yellow.
Because the Related works were merged into Introduction, the new numbering of sections were listed as follows:
1. Introduction
2. Methodology
2.1 Data Acquisition
2.2 Data Preprocessing
2.3 Feature Extraction
2.4 Physical Activity Classification
2.5 EE Prediction
3. Results
3.1 EE Estimation Performance of the General Model
3.2 Accuracy of Physical Activity Classification
3.3 EE Estimation Performance with Physical Activity Classification
4. Discussion
5. Conclusion

Reviewer 2 Report
This study sought to develop and validate a solution based on depth-camera and classification modelling able to estimate energy expenditure during different physical activities. The authors in particular used a validation setup with 3 depth cameras (i.e. Microsoft XBOX 360 Kinect cameras) at different locations (with side, rear side and rear views) with respect to the subjects performing different types of activity; furthermore, 3 estimation models (i.e., linear regression, multilayer perceptron and convolutional neural network) were used and compared in term of estimation performance. To train the model the authors used the 3D components of the velocity reached by 6 specific joints during the realization of the tasks, i.e., shoulder, elbow, wrist, hip, knee, and ankle of the left side; PCA was also considered to reduce eventually the feature dimensionality. Before estimating energy expenditure, the authors implemented a SVM classification model to classify the different type of activity (I.e., standing, walking and running). Focusing on a single camera use, the best performance was a achieved by the side view implementing the multilayer perceptron model, whereas – if better accuracy is required – the authors underlined the necessity to introduce a second camera and the convolutional neural network worked better for light-to-moderate activities. The authors claimed that this is the first study which proved the feasibility of this approach by considering different setup and estimation models for energy expenditure.
General Comment
This paper sought to add some information to a widely used application of depth-cameras for estimating energy expenditure. The problem is indeed widely investigated, although systematic analysis of camera location and estimation models has not yet been performed. As reported by the authors, this paper can be seen an extension of a previous work (i.e., Lin, B.‐S.; Wang, L.‐Y.; Hwang, Y.‐T.; Chiang, P.‐Y.; Chou, W.‐J. Depth‐Camera‐Based System for Estimating Energy Expenditure of Physical Activities in Gyms. IEEE J Biomed Health Inform. 2019, 23, 1086–1095, doi:10.1109/JBHI.2018.2840834).
The structure of the article is in general quite correct, including Introduction and Related Works, Methodology [with subheading], Results [with subheading], Discussion and Conclusions.
Methodologies and performance analysis seem to be correctly reported but further details are needed; several concerns are here reported to the authors.
The use of the English language seems to be correct for a non-native English speaker.
Title
I guess you are not classifying posture but different motor tasks; furthermore, I would underline that you used also different models to estimate energy expenditure based on kinematics acquired by depth-cameras. Indeed, this is not a presentation of a single system, but a quite wide validation study of different solutions.
Abstract
Ok. In general, well written. I would only add the information related to the model you used for task classification.
Introduction
In general, ok. Few notes:
- Line 39-49: For a general introduction about the need for physical activities, this section is ok. On the other hand, your system can not be widely and freely used in different real-life scenarios, like the smartwatch/wrist bands can. Your approach can be considered within an application field that is quite similar to that one you underlined, i.e., the COSMED system. So, it is a problem that is quite different with respect to the need for correct physical activities, that we are all performing in our daily life, but not within a laboratory or during outpatient assessment.
- Line 45: COSMED K4b2 is a commercial name for a general system that can be called; I would rather use the general name and then, eventually, provide some commercial examples.
- Line 49-53: I would avoid providing description of each section of the paper.
- Line 54: I would avoid using a different chapter for the analysis of the literature; I would put the information provided in this chapter together with the introduction.
- Line 56: Were all these previous studies you analyzed based on the only K4B2 system?
- At the end of the Introduction section, besides the main goal of your work, I would also define the main hypotheses.
Materials and Methods
In general ok.
- Line 147-148: Please justify the number of subjects you involved in your study.
- Table I. I guess this table report too much detail. Demographic data are fundamental, but in this case, I guess the synthetic information you reported are enough (check only the useful number of digits and the units of measurement).
- Line 157-158: Please justify the use of these values of speed for walking and running. Define the terms “light”, “moderate” and “vigorous” with respect to this information.
- Line 159-160: I guess you have to use a treadmill, since in free-living conditions you cannot follow the subjects with the depth-camera, unless you make the camera automatically movable.
- Line 180-189: I am not able to fully get the meaning of this transformation, which is a pure translation considering the direction of the axis concordant with the system reference frame. In this way, the displacements of the other joint centers are all referred relatively to the shoulder center and even the velocity you estimated is a relative velocity. Please explain better your choice.
- Line 192-196: Could you provide information about Kinect sampling rate as well, thus to understand the size of your window in time.
- Line 234-239: Although you reported the reference [22] please provide further few detail about SVM classifier.
- Line 245-249: We need further details about the structure of the MLP model; could you please provide a figure like Figure 6.
- Figure 5 and line250-255: This paragraph should be shifted in the Results section.
- Line 259-264: Even here in this paragraph, to fully understand the implementation of the CNN we need further information; it is not enough reporting 1@18x3 etc. in figure 6. Please provide full description in the main text. Furthermore, CNN is not that common in time series; please provide few more information about your choice.
Results
- Line 282-285: It is not clear what is this “General Model”. It is a model able to estimate the energy expenditure regardless of the performed task, isn’t it? Please explain it better.
- Table 3-5 (Table III-V in the main text, please check for coherence): Please provide the units of measurement.
- Table 6: Please provide information about the type of validation you used to obtain these values of accuracy (even better it should be explained in the Methodologies section).
- Line 316-318: This information should be placed in the Methodology section, as well.
Discussion
- Line 353-360: I guess you should underline also that the accuracy can be lower, but you reduced the complexity of the classification problem. You should analyze the impact of this reduction on a computational costs perspective. Is 1% lower justifiable?
- In the Discussion section you should underline also the main limitation of your study.
Conclusion
Ok. Well written.
References
Literature is extended and up-to-date.
Figures
Ok.
Table
See previous comments.
Author Response
Author’s replies to the comments of Reviewer 2:
[Comment 1]
This study sought to develop and validate a solution based on depth-camera and classification modelling able to estimate energy expenditure during different physical activities. The authors in particular used a validation setup with 3 depth cameras (i.e. Microsoft XBOX 360 Kinect cameras) at different locations (with side, rear side and rear views) with respect to the subjects performing different types of activity; furthermore, 3 estimation models (i.e., linear regression, multilayer perceptron and convolutional neural network) were used and compared in term of estimation performance. To train the model the authors used the 3D components of the velocity reached by 6 specific joints during the realization of the tasks, i.e., shoulder, elbow, wrist, hip, knee, and ankle of the left side; PCA was also considered to reduce eventually the feature dimensionality. Before estimating energy expenditure, the authors implemented a SVM classification model to classify the different type of activity (I.e., standing, walking and running). Focusing on a single camera use, the best performance was a achieved by the side view implementing the multilayer perceptron model, whereas – if better accuracy is required – the authors underlined the necessity to introduce a second camera and the convolutional neural network worked better for light-to-moderate activities. The authors claimed that this is the first study which proved the feasibility of this approach by considering different setup and estimation models for energy expenditure.
Answer:
Many thanks for the reviewer’s comments.
[Comment 2]
General Comment
This paper sought to add some information to a widely used application of depth-cameras for estimating energy expenditure. The problem is indeed widely investigated, although systematic analysis of camera location and estimation models has not yet been performed. As reported by the authors, this paper can be seen an extension of a previous work (i.e., Lin, B.‐S.; Wang, L.‐Y.; Hwang, Y.‐T.; Chiang, P.‐Y.; Chou, W.‐J. Depth‐Camera‐Based System for Estimating Energy Expenditure of Physical Activities in Gyms. IEEE J Biomed Health Inform. 2019, 23, 1086–1095, doi:10.1109/JBHI.2018.2840834).
The structure of the article is in general quite correct, including Introduction and Related Works, Methodology [with subheading], Results [with subheading], Discussion and Conclusions.
Methodologies and performance analysis seem to be correctly reported but further details are needed; several concerns are here reported to the authors.
The use of the English language seems to be correct for a non-native English speaker.
Answer:
Many thanks for the reviewer’s comments.
[Comment 3]
Title
I guess you are not classifying posture but different motor tasks; furthermore, I would underline that you used also different models to estimate energy expenditure based on kinematics acquired by depth-cameras. Indeed, this is not a presentation of a single system, but a quite wide validation study of different solutions.
Answer:
Many thanks for the reviewer’s comments. Yes, we classified different motor tasks and use different model to estimate the energy expenditure based on kinematics acquired by depth-cameras. We also evaluate different solutions, such as posture classification algorithm, to obtain the optimal solution for different type of physical activity and camera settings.
[Comment 4]
Abstract
Ok. In general, well written. I would only add the information related to the model you used for task classification.
Answer:
Many thanks for the reviewer’s comments. We have added the information of the model for physical activity classification in line 23 to 24 in the Abstract.
[Comment 5]
Introduction
In general, ok. Few notes:
Line 39-49: For a general introduction about the need for physical activities, this section is ok. On the other hand, your system can not be widely and freely used in different real-life scenarios, like the smartwatch/wrist bands can. Your approach can be considered within an application field that is quite similar to that one you underlined, i.e., the COSMED system. So, it is a problem that is quite different with respect to the need for correct physical activities, that we are all performing in our daily life, but not within a laboratory or during outpatient assessment.
Line 45: COSMED K4b2 is a commercial name for a general system that can be called; I would rather use the general name and then, eventually, provide some commercial examples.
Line 49-53: I would avoid providing description of each section of the paper.
Line 54: I would avoid using a different chapter for the analysis of the literature; I would put the information provided in this chapter together with the introduction.
Line 56: Were all these previous studies you analyzed based on the only K4B2 system?
At the end of the Introduction section, besides the main goal of your work, I would also define the main hypotheses.
Answer:
Many thanks for the reviewer’s comments. The revisions of Introduction are listed as follows. All of the modified parts were also highlighted in the manuscript.
- The proposed system is mainly for indoor physical activities. The motivation is that some elderly can not go outside for exercising, but it is still important for them to meet the basic physical activities requirement. We have added this information in line 45 to 48 in Introduction.
- The general name of COSMED K4b2 is portable metabolic analyzer. In addition to COSMED K4b2, there are other commercial examples of portable metabolic analyzer like PNOĒ and METAMAX® We have modified the COSMED K4b2 to portable metabolic analyzer and provided the commercial examples in line 49 to 53 in Introduction.
- We have removed the sentences of description of each section of this paper.
- We have merged the Related Works into Introduction and modified all of the numbering of the sections of this article.
- Most of the systems used K4b2, but some of them used another brand of metabolic analyzer, such as COSMED Quark CPET and Vyntus™ CPX Metabolic Cart, etc. We modified the term “K4b2 system” in line 57 of related works to “metabolic analyzers” because not all of the systems used K4b2 system as the ground truth.
- We have added the main hypotheses of the work “The proposed system hypothesized that using posture classification algorithm can improve the accuracy of EE estimation and expected to obtain the optimal model for each physical activity for different camera settings.” at the end of the Introduction section and highlighted it.
[Comment 6]
Materials and Methods
In general ok.
Line 147-148: Please justify the number of subjects you involved in your study.
Table I. I guess this table report too much detail. Demographic data are fundamental, but in this case, I guess the synthetic information you reported are enough (check only the useful number of digits and the units of measurement).
Line 157-158: Please justify the use of these values of speed for walking and running. Define the terms “light”, “moderate” and “vigorous” with respect to this information.
Line 159-160: I guess you have to use a treadmill, since in free-living conditions you cannot follow the subjects with the depth-camera, unless you make the camera automatically movable.
Line 180-189: I am not able to fully get the meaning of this transformation, which is a pure translation considering the direction of the axis concordant with the system reference frame. In this way, the displacements of the other joint centers are all referred relatively to the shoulder center and even the velocity you estimated is a relative velocity. Please explain better your choice.
Line 192-196: Could you provide information about Kinect sampling rate as well, thus to understand the size of your window in time.
Line 234-239: Although you reported the reference [22] please provide further few detail about SVM classifier.
Line 245-249: We need further details about the structure of the MLP model; could you please provide a figure like Figure 6.
Figure 5 and line250-255: This paragraph should be shifted in the Results section.
Line 259-264: Even here in this paragraph, to fully understand the implementation of the CNN we need further information; it is not enough reporting 1@18x3 etc. in figure 6. Please provide full description in the main text. Furthermore, CNN is not that common in time series; please provide few more information about your choice.
Answer:
Many thanks for the reviewer’s comments. The revisions of Materials and Methods are listed as follows. All of the modified parts were also highlighted in the manuscript.
- We have added the number of subjects in the study. 21 subjects were recruited in this study.
- We have modified Table 1 by removing the details of subjects and reporting only the synthetic information of the 21 subjects. The revised version of Table 1 was highlighted.
- The light, moderate, and vigorous activities in this study are standing, walking, and running. We have added this information in line 160 to 161 and Table 2.
- The reason of using treadmill in this study is that it is much easier to stabilize the speed of each activities and compare with other related studies.
- According to page 5 in ref [15] in the manuscript, it is necessary to shift the origin of coordinate from Kinect to a point in the object body to solve the action recognition program caused by different object distances from Kinect. Therefore, before using the kinematics from Kinect, all of the values should be translated to a coordinate system which its origin is the point on the human body rather than Kinect camera. After doing the transformation, the distance factor between the object and Kinect will be neutralized. Shoulder center was used as the origin of the body center in this reference so we decide to use shoulder center as the referenced origin point. The figures before and after the coordinate transformation are presented as follows. We have added more information about the reason of using the translation in line 185 to 191 in section 2.2.
|
(a) Before coordinate transformation |
|
|
|
(b) After coordinate transformation |
- The sampling rate of Kinect is 30 Hz. We have added this information at the end of paragraph 1, section 2.1.
- We have added more information about SVM in line 244 to 248 in section 2.4.
- We have added the figure of MLP as Figure 5. The hidden neurons for each layer were remained n in this Figure because the optimal hidden neurons will be shown in the Result section.
- We have moved Figure 5 to Result section and relabeled it as Figure 7. We remained the method for evaluating the optimal number of neurons in section 2.5 and moved the description of the result to Result section.
- We have added more information of the CNN model in line 276 to 280 in section 2.5. The reason why we chose the CNN is that we would like to detect the features automatically. Some studies also indicated that there are some benefits to use CNN rather than recurrent neural network (RNN) for time-series data. For example, the computational cost of CNN is less than RNN because CNN uses batch learning but RNN trains the data sequentially. CNN also doesn’t have any assumption of the previous data and can see data from the broader perspective. We have added the reason why we chose CNN for the dataset in line 271 to 272 in section 2.5.
[Comment 7]
Results
Line 282-285: It is not clear what is this “General Model”. It is a model able to estimate the energy expenditure regardless of the performed task, isn’t it? Please explain it better.
Table 3-5 (Table III-V in the main text, please check for coherence): Please provide the units of measurement.
Table 6: Please provide information about the type of validation you used to obtain these values of accuracy (even better it should be explained in the Methodologies section).
Line 316-318: This information should be placed in the Methodology section, as well.
Answer:
Many thanks for the reviewer’s comments. The revisions of Results are listed as follows. All of the modified parts were also highlighted in the manuscript.
- Yes, the general model means the model can estimate the energy expenditure regardless of the performed task. We have modified the sentence in line 310 to 311 to make it clearer.
- The units for MAE, MSE, and RMSE are MET, MET2, and MET, respectively. We had added the units in Tables 3 to 5, and Tables 7 to 9. We also checked the coherence of numberings of Tables and main text and fixed all of the mistakes.
- The accuracy in Table 6 was calculated from the average of 10-fold cross validation.
- We have changed the term “K-fold” to “10-fold” and moved the three sentences to the last paragraph in section 2.5.
[Comment 8]
Discussion
Line 353-360: I guess you should underline also that the accuracy can be lower, but you reduced the complexity of the classification problem. You should analyze the impact of this reduction on a computational costs perspective. Is 1% lower justifiable?
In the Discussion section you should underline also the main limitation of your study.
Answer:
Many thanks for the reviewer’s comments. The revisions of Discussion are listed as follows. All of the modified parts were also highlighted in the manuscript.
- We have added the description to underline that although the accuracy of using PCA is lower compared to not using PCA in this study, using PCA might still be a practical approach for analyzing the larger dataset because the reduction effect of the larger dataset would be more significant than that of the smaller dataset. 1% lower is justifiable because after using PCA, the cumulative explained variance is 0.9 which means that some characteristic would be dropped. The information was added in line 380 to 383.
- We have moved the descriptions of limitations from Conclusion to the last two paragraphs of Discussion.
[Comment 9]
Conclusion
Ok. Well written.
References
Literature is extended and up-to-date.
Figures
Ok.
Table
See previous comments.
Answer:
Many thanks for the reviewer’s comments.
